# Exploring Relevant mRNAs and miRNAs in Injured Urethral Tissues of Rats with High-Throughput Sequencing

**DOI:** 10.3390/genes13050824

**Published:** 2022-05-05

**Authors:** Han Lin, Shiyong Guo, Song Li, Jihong Shen, Jianfeng He, Yun Zheng, Zhenhua Gao

**Affiliations:** 1Department of Urology, The First Affiliated Hospital of Kunming Medical University, Kunming 650032, China; linhan912.cn@gmail.com (H.L.); ls198759@126.com (S.L.); kmsjh99@aliyun.com (J.S.); 2State Key Laboratory of Primate Biomedical Research, Institute of Primate Translational Medicine, Kunming University of Science and Technology, Kunming 650500, China; shiyongguo_001@163.com; 3Faculty of Information Engineering and Automation, Kunming University of Science and Technology, Kunming 650500, China; jfenghe@kust.edu.cn

**Keywords:** RNA-seq, sRNA-seq, urethral injury, mRNA, miRNA, inflammation

## Abstract

Acute urethral injuries caused by urethral endoscopy and other mechanical injuries are the main reasons for secondary infection and late urethral stricture. However, there are no studies to explore the transcriptomic changes in urethral injury and the molecular mechanism of urethral injury, which is important for the treatment and cure of urethral injury. Therefore, we used RNA-seq and sRNA-seq profiles from normal and injured urethral tissues to identify and characterize differentially expressed mRNAs and miRNAs. In total, we found 166 differentially expressed mRNAs, of which 69 were upregulated, and 97 were downregulated in injured urethral tissues. The differentially expressed mRNAs were mainly involved in the positive regulation of epithelial cell differentiation, focal adhesion, cell adhesion molecules, protein activation cascade, complement activation, complement and coagulation cascades, and chemokine-mediated signaling pathway. Additionally, we found six upregulated and four downregulated miRNAs, respectively, in the injured urethral tissues. Notably, their target genes were involved in the vascular endothelial growth factor receptor 2 binding, PI3k-Akt signaling pathway, and Notch signaling pathway. In summary, our results suggest that the cell damage response induced by mechanical injury activates the pathological immune response in a variety of ways in injured urethral tissues.

## 1. Introduction

The human urethra is a natural cavity directly connected with the outside world. It is the originator of endoscopy and endoluminal surgery. However, with the development of minimally invasive treatment methods such as endoscopic surgery and robotic surgery, as well as the rapid development of physical therapy methods such as laser, plasma, and radiation, patients are inevitably prone to iatrogenic urethral injury while enjoying the advantages of minimally invasive treatment [1]. Although urethral injury itself is not life-threatening, if it is not accurately diagnosed and reasonably treated, most urethral injuries will turn into urethral stricture [2,3]. Once it occurs, patients must receive regular endoscopic intervention, urethral dilation, and/or cleaning intermittent catheters, which have been used for many years. In more advanced and recurrent stenosis, open end-to-end anastomosis and/or free transplantation are required, and long-term rehabilitation, including follow-up surgery [4,5,6], is required, which imposes severe psychological and financial burdens on the patient and seriously affects the patient’s quality of life. More than 12,000 patients in the United Kingdom need surgery for urethral stricture every year, which costs about 10 million pounds a year [7]. In the United States, there have been more than 5000 hospitalized patients with urethral stricture, with an annual cost of 200 million dollars [8]. China’s population is about four times that of the United States and 21 times that of the United Kingdom, suggesting that the number of people suffering from urethral stricture is much larger and the economic burden is more intensive in China [9]. Therefore, exploring the relevant molecular mechanisms that occur in urethral injury tissue is particularly important for preventing and/or slowing down urethral stricture.

Previous studies have shown that after urethral injury, local tissues and cells undergo degeneration and necrosis, vascular permeability increases, and leukocyte exudation [10,11]. Subsequently, lysosomal enzymes, active oxygen free radicals, prostaglandins, and a variety of inflammatory mediators [12,13,14] are released to mediate vascular endothelial cell and tissue damage. These promote the infiltration of lymphocytes, macrophages, and plasma cells, resulting in tissue damage response and tissue repair [11]. At the same time, fibroblasts activate and proliferate, resulting in the increase in synthetic collagen fibers [15], which will reduce the extension and compliance of the urethral cavernous body and finally lead to the formation of urethral lumen stenosis [16]. In summary, urethral stenosis is a complex pathological process from urethral injury to urethral stenosis, which involves a series of interlocking molecular mechanisms. Feng et al. [17] found that urethral injury leads to the activation of the TGF-β1 signal, which further promotes the proliferation, activation, and migration of urethral fibroblasts reduces the secretion of IP10 by fibroblasts, inhibits the IP10/CXCR3 signaling pathway, accelerates the pathological process of urethral fibrosis, and ultimately leads to urethral stricture. Although multiple mechanisms of urethral injury to urethral stricture have been proposed, the analysis of the molecular signaling events involved remains incomplete.

High-throughput sequencing technology, also known as next-generation sequencing technology, is a milestone in the history of sequencing. It can simultaneously sequence millions of DNA or RNA. After decades of development, it has been widely used in the genome, including sequencing and epigenomics, as well as many aspects of functional genomics research [18].

In the past few decades, non-coding RNAs have been considered transcriptional noise because they do not encode proteins. Studies have revealed that non-coding RNAs are involved in many pathophysiological processes [19]. miRNAs are a kind of non-coding single-stranded RNAs with a length of ~22 nucleotides. It is transcribed from DNA and is not translated into protein. It is mainly involved in the expression regulation of post-transcribed genes [20]. Studies have shown that miRNA induces gene expression silencing by binding to their target mRNAs, blocking translation initiation, extension, or by including the degradation of mRNAs [21]. Because miRNAs do not need to be completely complementary to inhibit gene expression, a specific miRNA can regulate multiple gene transcripts, and a specific gene transcript can also be inhibited by multiple miRNAs [21]. miRNA is involved in the pathophysiological process of a variety of traumatic diseases, such as radiation-induced esophageal injury [22], acute kidney injury [23], myocardial injury [24], and traumatic spinal cord injury [25].

The molecular mechanism of the role of mRNA and miRNA in the occurrence and development of urethral injury is still unclear. In order to study the gene regulation mechanism that occurs in urethral injury tissues, we used high-throughput sequencing technology to study the differential expression of mRNA and miRNA in urethral injury tissues. Then we predicted the biological functions of differentially expressed mRNA and miRNA target genes using Gene Ontology (GO) enrichment analysis, Kyoto Encyclopedia of Genes and Genomes (KEGG) pathway analysis, and Gene Set Enrichment Analysis (GSEA). These results can provide new insights and potential intervention targets for early intervention or treatment of urethral stricture caused by urethral injury.

## 2. Materials and Methods

### 2.1. Animals

Four specific pathogen-free (SPF) male Sprague Dawley (SD) rats (250–300 g) aged 6–8 weeks old were used as the research objects. They were randomly divided into an injured group and a normal group, with 2 rats in each group. All rats were purchased from the Department of Laboratory Animal Science, Kunming Medical University (Kunming, China), (Animal Quality Certificate Number: SCXK (Dian) K2020-0004). The SD rats of the injured and normal groups were housed in two cages separately. All rats were housed in an SPF barrier environment (experimental facility license number: SYXK (Dian) K2020-0006) using a 12 h light–dark cycle and had free access to food and water with a temperature range of 20 °C–25 °C, relative humidity range of 50% to 70%. All procedures had been approved by the Experimental Animal Ethics Committee of Kunming Medical University (approval number: kmmu2021721).

### 2.2. Establishment of Urethral Injury Animal Model

The rats in the injured group were anesthetized with an intraperitoneal injection of pentobarbital sodium (50 mg/kg). After the rats were successfully anesthetized, they were fixed on the small animal constant temperature pad on the operating table in a supine position to expose the operation area, and the rat hair in the perineum was removed with a rechargeable shaver. The perineal area was carefully disinfected with iodophor, and urinary catheterization was performed. Then the abdomen was pressed to empty the urine from the bladder as much as possible. After pulling out the urinary catheter, an elastic tourniquet was placed at the root of the penis to reduce intraoperative bleeding. The skin of the ventral penis was cut to expose the urethra, and a small opening was cut at the urethral orifice with ophthalmic scissors. The urethral orifice was pulled open with three vascular forceps, and the urethra was slightly expanded with a specific round end iron bar. Then, the round end of an iron bar was baked on the outer flame of the alcohol lamp for 6–7 s and inserted into the urethra to make an insertion depth of about 26 mm. After each withdrawal, the bar was baked for 6–7 s and inserted again with 2–3 repetitions to scald the urethral mucosa. Then, the tourniquet was removed, and the skin of the penis was sutured with 5-0 absorbable thread. After waking up, the rats were placed in the feeding cage with a sterilized clean pad and placed in the feeding room. Drinking water and feed were sufficient, and the rats were free to eat and drink water after surgery. The life activity status and urethral orifice of the rats were closely observed. The surgical site of the injured group was disinfected 3 days after the operation.

### 2.3. Tissue Collection

On the 3rd day after modeling the urethral injury, rats were euthanized by intraperitoneal injection of sodium pentobarbital (150–200 mg/kg). After checking for cardiac arrest, we then fixed the rats on the operating table in a supine position to expose the operation area. We removed the rat hair in the perineum with a rechargeable shaving device and cut the skin of the ventral penis to expose the urethra. We completely removed the urethra from the lower curve of the pubis to the distal end of the urethra with ophthalmic scissors and placed it on ice. We carefully and quickly removed the surrounding excess tissue and then weighed the samples and put it into the labeled enzyme-free cryopreservation tube and soaked them in liquid nitrogen.

### 2.4. RNA Isolation, Library Preparation and Sequencing

Each sample contained at least 60 mg of urethral tissues for RNA extraction. In accordance with instructions provided by the manufacturer, total RNA was extracted from the tissues using Trizol (Invitrogen, Carlsbad, CA, USA). Subsequently, total RNA was qualified and quantified using a Fragment Analyzer instrument (Agilent Technologies, Santa Clara, CA, USA). For all samples, RNA integrity number values were greater than 7. RNA-seq and sRNA-seq libraries were constructed, followed by cluster formation and sequencing using the BGISEQ-500 platform (BGI-Shenzhen, Shenzhen, China).

### 2.5. Generation and Analysis of the RNA-seq Profiles in Rat Urethra

The BGISEQ-500 sequencer was used with a 2 × 150 pair-end RNA-seq strategy to sequence the total RNAs of the four samples. We deposited the obtained sequencing data into the NCBI GEO database under the accession number GSE182642. The genome and annotation of rat were downloaded from UCSC (http://genome.ucsc.edu (accessed on 12 September 2021)). The alignment was performed using the options of “-p 24 -dta-cufflinks -q -S” within HISAT2 (v 2.1.0) [26]. StringTie (v 2.1.6) [27] was used to assemble the transcriptomes with the options of “-p 20 -l normal_2A”. Moreover, cuffquant and cuffnorm in the Cufflinks package (v 2.2.1) [28] were used to calculate and normalize the expression levels of transcripts, respectively. Using the edgeR program (v 3.34.0) [29], we compared the expression levels of transcripts in normal and injured urethral tissues in rats. A total of 25,718 transcripts were obtained after filtering the expression levels of transcripts with mean expression levels of at least 1 FPKM (Fragments Per Kilo basepairs per Million sequencing reads) in two tissues of comparisons. The FPKM values of the transcripts were compared with the edgeR program (v 3.34.0) [29]. The transcripts with corrected *p*-values smaller than 0.05 and |log_2_(Fold Change (FC))| ≥ 1.5 were considered statistically significant. KOBAS (v 3.0) [30] was used to obtain enriched GO terms and KEGG pathways based on these differentially expressed transcripts. The GO analysis exported three categories: biological process (BP), cellular component (CC), and molecular function (MF). The *p*-values were corrected by the Benjamini–Hochberg procedure, and GO terms and KEGG pathways with corrected *p*-values < 0.05 were considered significantly enriched. A bi-clustering analysis of the expression levels of filtered transcripts was carried out using the pheatmap package in R.

### 2.6. Gene Set Enrichment Analysis

We used HISAT2 (v 2.1.0) [26] to align the obtained RNA-seq paired-end reads to the rat genome. Samtools (v 1.12) [31] was used to convert files from sam format to bam format using the options of “view -bS -T” and sort reads by name with the options of “sort”. FeatureCounts (v 2.0.3) [32] was used to calculate the expression levels of genes in each sample with the options of “-T 24 -p -t -g -a”. The GenomicFeatures (v 1.44.0) [33] package in R was used to convert counts to FPKM values of genes in each sample. Finally, gene set enrichment analysis was conducted based on the Gene Ontology and KEGG databases using the Gene Set Enrichment Analysis (GSEA) (v 4.1.0) [34] software. The expression data with a total of 32,883 normalized genes were uploaded to the GSEA software ((v 4.1.0), Broad Institute, Cambridge, MA, USA). Gene set size filters (min = 15, max = 500) resulted in filtering out 10,084/15,760 gene sets. The remaining 5676 gene sets were used in the analysis. The default weighted enrichment statistic was adopted to process the data 1000 times, and probes were ranked by the signal-to-noise ratio. A gene set was considered significantly enriched when the *p*-value was less than 0.05 and the false discovery rate (FDR) was less than 0.25. ES (enrichment score) reflects the degree to which a gene set is overrepresented in our dataset of differentially expressed genes. NES (normalized enrichment score) is the ES normalized for the gene set size.

### 2.7. Small RNA Sequencing Profiles

At the sequencing facility, small RNAs were isolated from total RNAs and used to prepare small RNA sequencing libraries. The obtained sRNA-seq libraries were sequenced by a BGISEQ-500 sequencer. Our sRNA-seq profiles were deposited in the NCBI GEO database with the accession number GSE182947. The FASTQC program (https://www.bioinformatics.babraham.ac.uk/projects/fastqc/ (accessed on 1 August 2021)) was used to examine the qualities of the obtained small RNA sequencing profiles. Computational analysis of sRNA reads obtained from the sRNA libraries was performed as reported previously [35]. At first, reads with low-quality scores (<30) of the first 25 nt and shorter than 18 nt were discarded. The remaining reads used the pipeline proposed previously [35] to generate an sRNA tissue frequency file.

*Rattus norvegicus* mature miRNA sequences were downloaded from the miRBase ((v 22.1), accessed on 12 September 2021) [36], and the unique mature miRNA sequences were obtained. Finally, the frequencies of mature miRNAs in different small RNA sequencing profiles were calculated by aligning the sRNA tissue frequency file to unique mature miRNAs using NCBI BLASTN (v 2.2.26) [37] with the options of “-S 1 -m 8 -e 0.01” and normalized to Reads Per Ten Million (RPTM) sequencing tags. In all, 526 miRNAs were identified after filtering expression levels of miRNAs that had at least one RPTM in two tissues of comparison. The RPTM values of miRNAs were compared with the edgeR program (v 3.34.0) [29]. miRNAs with *p*-values smaller than 0.05 and |log_2_(Fold Change (FC))| ≥ 0.5 were considered statistically significant. By using the pheatmap package in R, the bi-clustering analysis of the expression levels of filtered miRNAs was performed.

### 2.8. Identification of miRNA Targets and GO and KEGG Enrichment Analysis

To better understand the functional ramifications of the identified miRNAs, we predicted the miRNA targets with the HitSensor ((Release 4), Shanghai, China) [38] pipeline. The HitSensor algorithm searches miRNA complementary sites in coding regions with a modified Smith-Waterman algorithm [39]. It scores these sites by giving rewards to key sequence-specific determinants, including seed region, 12–17 nt region, local-AU content around the seed region, and ≤3 mismatches. After calculating the scores of the 5 determinants for each alignment site, it will add these individual determinant contributions to the alignment score to find the total score of a miRNA complementary site. Finally, sites with total scores larger than 472 are outputted. Because miRNAs normally repress the expression of their targets in a mild way [21], we choose targets whose expression levels show opposite change to the expression change of miRNAs. By using KOBAS (v 3.0) [30], enriched GO terms and KEGG pathways were obtained for the miRNA targets. We used Fisher’s exact test to find the *p*-values. GO terms and KEGG pathways with *p*-values < 0.05 were considered significantly enriched. miRNA:target relations were visualized using Cytoscape (v3.8.2) [40].

## 3. Results

### 3.1. Quality Examination of RNA-seq Data

To obtain expression patterns of genes in different conditions of rat urethra, normal and injured urethral tissues of male SD rats were collected. The total RNA from the four samples was isolated and used to prepare Ribo (-) RNA-seq libraries, which were then sequenced on a BGISEQ-500 sequencer (BGI-Shenzhen, Shenzhen, China). The sequencing quality of these RNA-seq libraries was examined by FASTQC (https://www.bioinformatics.babraham.ac.uk/projects/fastqc/ (accessed on 1 August 2021)) (Appendix A). A total of almost 40 million clean reads were obtained for each of these four RNA-seq profiles (Appendix A). Moreover, these samples have the average GC content of the reads of 50% and 51% in the normal and injured urethral tissue groups, respectively. The FASTQC results indicated that the sequencing clean data were of good quality, which was crucial to the next step of the analysis. HISAT2 was then used to align the sequencing reads to a rat genome reference. In total, at least one end of the 72,955,661 (90.2%) and 73,348,269 (90.6%) reads for the normal and injured urethral tissues, respectively, was successfully mapped back to the reference genome, indicating the good quality of the sequencing libraries as well as excellent credibility of the results in the downstream analysis.

### 3.2. Gene Expression Patterns of Injured Rat Urethra

The RNA-seq profiles were aligned to the rat genome, which was downloaded from UCSC (http://genome.ucsc.edu/ (accessed on 12 September 2021)) with HISAT2 [26]. Following alignment, transcriptomes were assembled, and gene abundances were calculated using StringTie [27] and Cufflinks [28], respectively (Appendix A). After filtering out lowly expressed transcripts, a total of 25,718 genes with mean expression levels of at least 1 FPKM in two tissues of comparisons were used in further analysis. The edgeR program [29] was used to identify the differently expressed genes (DEGs) in the injured rat urethra. A total of 166 differentially expressed genes in injured tissues were identified when compared with normal tissues (Appendix A). Among these DEGs, 69 were found to be upregulated while 97 were downregulated (Figure 1a). In the bi-clustering analysis, gene expression profiles from the same groups were clustered together using the DEGs, as shown in Figure 1b.

### 3.3. Functional Enrichment Analysis of Differentially Expressed Genes

Using these DEGs, we applied KOBAS (v 3.0) [30] to identify enriched GO terms and KEGG pathways in different groups (Appendix A, respectively). The results revealed that the upregulated genes in injured tissues were associated with negative regulation of cytosolic calcium ion concentration, the release of sequestered calcium ion into the cytosol by the sarcoplasmic reticulum, positive regulation of epithelial cell differentiation, positive regulation of membrane protein ectodomain proteolysis, inner cell mass cell proliferation, extracellular space, stress fiber, actin filament binding, lipoteichoic acid binding, peptidase activator activity, and lipopolysaccharide binding (Figure 2a). On the other hand, downregulated genes in injured tissues mainly consisted of the GO terms, such as the synaptic signaling, negative regulation of endopeptidase activity, regulation of sensory perception of pain, retina homeostasis, kinocilium, extracellular matrix, endopeptidase inhibitor activity, small molecule binding, and WW domain binding and scavenger receptor activity (Figure 2c). The majority of the upregulated genes in injured tissues belong to the phagosome, focal adhesion, ECM-receptor interaction, cellular senescence, and cell-adhesion molecule pathways (Figure 2b), which are closely related to inflammatory response, tissue damage, and repair. However, downregulated genes in injured tissues were associated with the tight junction, salivary secretion, endocytosis, and allograft rejection pathways, including graft-versus-host disease with the highest Rich Factor (Figure 2d).

### 3.4. The GSEA Analysis

To study the molecular mechanism related to injury of the urethra tissues, we performed the gene set enrichment analysis using the GSEA [34] software (Appendix A, respectively). The GSEA analysis showed that the gene sets of the protein activation cascade, complement activation, antigen processing and the presentation of peptide or polysaccharide antigens via MHC class II complement and coagulation cascades, positive regulation of vascular endothelial growth factor production, phagocytosis, engulfment, chemokine-mediated signaling pathway, cellular response to interferon-γ, and monocyte chemotaxis were significantly enriched in the injured group (Figure 3). These results were highly related to the activation of an inflammatory response, the release of inflammatory factors, tissue damage, and repair. On the other hand, keratin filament, oxidative phosphorylation, mitochondrial protein complex, respiratory chain complex, inner mitochondrial membrane protein complex, and the respiratory electron transport chain were the most relevant gene sets to the normal group (Appendix A). This indicated that there were more biological processes related to mitochondrial activity in normal tissues, which seemed to mean that mitochondria were involved in the maintenance of the normal tissues.

### 3.5. Small RNA Sequencing Profiles of Different Tissues of Rat Urethra

Using a BGISEQ-500 sequencer, we obtained four small RNA-seq (sRNA-seq) profiles from the same samples of rat urethra that were used for RNA-seq. The sequencing quality of these sRNA-seq reads was examined by FASTQC (Appendix A). After the removal of the low-quality reads and the 3′ adaptors, an average of 33 million clean reads with ≥18 nt for each of these four sRNA-seq profiles were obtained (Appendix A). In these four sRNA-seq libraries, we examined the length distributions of the small RNA reads and unique sequences. The peak at 22 nt was found in all libraries for reads, suggesting the good quality of these libraries (Appendix A).

### 3.6. Conserved miRNAs and Their Expression Patterns in Different Tissues

A pipeline proposed previously [35] was used to generate an sRNA tissue frequency file for the four sRNA-seq libraries. The frequencies of mature miRNAs in different small RNA sequencing profiles were then calculated by aligning the sRNA tissue frequency file to unique mature miRNAs available at the miRBase (v 22) [36] and normalized to Reads Per Ten Million (RPTM) sequencing tags. After filtering out lowly expressed miRNAs, a total of 526 miRNAs with mean expression levels of at least 1 RPTM in one of the groups were used in further analysis (Appendix A). The edgeR program [29] was used to identify differentially expressed miRNAs (DEMs) in different tissues of rat urethra. An analysis of these data revealed a total of 10 miRNAs that were differentially expressed in injured tissues (Appendix A). Among these DEMs, six were found to be upregulated, while four were downregulated (Figure 4a). The bi-clustering was performed for the DEMs. As shown in Figure 4b, samples of the same groups were clustered together in the clustering analysis of the miRNA expression profiles.

### 3.7. Target Gene Prediction and Functional Regulatory Network of DEMs

As miRNAs may regulate protein-coding genes, we further predicted the target genes of the identified DEMs using the HitSensor algorithm [38]. A large number of putative targets for conserved miRNAs were identified (Appendix A). Because miRNAs normally repress the expression of their targets in a mild way [21], we chose targets whose expression levels show opposite change to the expression change of miRNAs. By filtering these miRNAs: target pairs in the edgeR results of RNA-seq analysis, a total of six upregulated miRNAs corresponding to 59 downregulated genes (logFC < −0.19) and two downregulated miRNAs corresponding to 23 upregulated genes (logFC > 0.1) were used for further analysis (Appendix A). We thus obtained complex regulatory networks between miRNAs and their target genes, as illustrated in Figure 5a,b.

These target genes of DEMs were applied to KOBAS (v 3.0) [30] to identify the enriched GO terms and KEGG pathways (Appendix A, respectively). The results revealed that genes corresponding to upregulated miRNAs in injured tissues compared with normal tissues were associated with the vascular endothelial growth factor receptor-2 signaling pathway, positive chemotaxis, tube formation, regulation of p38MAPK cascade, cytoplasm, cytosol, vascular endothelial growth factor receptor 2 binding, platelet-derived growth factor receptor binding, and protein kinase binding (Figure 6a). On the other hand, genes corresponding to downregulated miRNAs in injured tissues mainly consisted of the GO terms, such as the cAMP response element binding protein binding, mRNA 3′-UTR binding, nucleus, positive regulation of cell adhesion molecule production, multicellular organismal reproductive process, and the regulation of mRNA stability (Figure 6c). The majority of the genes corresponding to the upregulated miRNAs in injured tissues belong to the PI3k-Akt signaling pathway, Apoptosis and Cell adhesion molecules (CAMs) pathways (Figure 6b). However, genes corresponding to downregulated miRNAs in injured tissues were associated with RNA transport, RNA degradation, Notch signaling pathway, Metabolic pathways, HIf-1 signaling pathway, and Glutathione metabolism pathways (Figure 6d).

## 4. Discussion

Using RNA-seq, we investigated how mechanical injury induced whole transcriptome alterations in injured and normal urethral tissues. We identified 166 differentially expressed genes between the injured and normal groups, including 69 upregulated and 97 downregulated genes. The functional annotation of injury-altered expression genes shows that the significantly upregulated genes included the inflammatory and immune response (*Ccl6*, *Casp1*, *Anxa2*, *Rnf7*, *Plaur*, *Anpep*, *LOC100911413*, *App*, *C4b*, *RT1-A1*, *Sptan1*, and *Tuba1a*), wound healing and tissue regeneration (*Plaur*, *LOC100911413*, *Lamb1*, *Col14a1*, and *Mylk*), and cell apoptosis (*Sptan1*, *Tuba1a*, *Ccl6*, *App*, *Igfbp3*, and *Casp1*). The aberrant expression of some of these genes was previously reported to be important in the development of inflammatory cascade. This evidence suggested that injured urethral tissue showed a heavy inflammatory response and coexistence of tissue regeneration, as described in previous studies concerning neuroprotection of traumatic brain injury [41]. However, *Pygm*, *Pdcd4*, *Atp5pf*, *Rpl5*, *Ca3*, *Adgrf5*, *Smarca2*, *Eef1a2*, *Aplp2*, and *Myh1*, which were involved in metabolic processes and ATP synthesis, were downregulated after injury, indicating that those mitochondrial functional associated genes might be highly sensitive to the injury of the urethral tissue. Zhou et al. [42] found that damaged mitochondria release reactive oxygen species, which further activate the NLRP3 inflammasome. NLRP3 inflammasome is a molecular complex that triggers innate immune defense through the maturation of pro-inflammatory cytokines such as interleukin-1beta (IL-1beta) in response to danger signals, such as from infection and metabolic disorders [43,44]. This was also consistent with our GSEA results, and we speculated that the clearance of damaged mitochondria might be a potential intervention target to deal with injury-induced inflammation.

Next, the functional enrichment analysis of the differentially expressed genes was performed by GO and KEGG pathway analysis. We found that the release of sequestered calcium ions into the cytosol by sarcoplasmic reticulum, positive regulation of epithelial cell differentiation, inner cell mass cell proliferation, peptidase activator activity, WW domain binding, phagosome, focal adhesion, and ECM-receptor interaction could be meaningfully related to the regulation of mechanical damage treatment in urethral tissues. Indeed, some of these GO terms and KEGG pathways have also been reported in esophageal injury induced by ionizing radiation [45].

Our GSEA results showed that the protein activation cascade, complement activation, complement and coagulation cascades, positive regulation of vascular endothelial growth factor production, phagocytosis, chemokine-mediated signaling pathway, cellular response to interferon-γ, and monocyte chemotaxis were the most significantly enriched gene sets in injured group. Complements serve as an initial defense mechanism against unwanted host elements or nonself cells [46]. Complement-mediated functions range from direct cell lysis to the regulation of humoral and adaptive immunity [47]. Moreover, this system also regulates many inflammatory and immunological processes [48]. Angiogenesis is a hallmark of wound healing, the menstrual cycle, cancer, and various ischemic and inflammatory diseases. Vascular endothelial growth factor (VEGF) is an interesting inducer of angiogenesis and lymphangiogenesis; it binds to tyrosine kinase receptors and results in endothelial cell proliferation, migration, and new vessel formation [49]. In our GSEA analysis, we found that the expression of complement *C5* (*C5*) was upregulated after injury. According to its functional annotation, we found that it was highly correlated with the production of vascular endothelial growth factors. In addition, it was also correlated with complement activation, inflammatory response, and regulation of chemotaxis. Sinno et al. [50] found that complement *C5* has been shown to be chemotactically active for monocytes and polymorphonuclear leukocytes. It strengthens the inflammatory response by promoting the release of free radicals and tissue digestive enzymes by inflammatory cells [51]. On the other hand, the increase in inflammatory cells stimulates epithelialization, fibroblast activation, and collagen deposition, which further increases the breaking strength of the wound [11,12]. In general, the results of GSEA suggested that in mechanically induced injured tissues, genes related to complement activation and angiogenesis were activated, leading to the occurrence of inflammatory response and accelerating tissue repair. We speculated that early activation of *C5* was helpful for tissue repair of mechanically induced injury.

Through sRNA-seq, we found that there were 10 differentially expressed miRNAs between normal and injured samples, of which six (rno-miR-212-5p, rno-miR-339-5p, rno-miR-31a-3p, rno-miR-34b-3p, rno-miR-532-3p, rno-miR-31a-5p) were upregulated and 4 (rno-miR-486, rno-miR-503-5p, rno-miR-376a-3p, rno-miR-410-3p) were downregulated in the injured samples. In our study, we found that *Tp53inp1* was targeted by one upregulated miRNA, miR-339-5p (Figure 5a,c). Tumor protein p53-inducible nuclear protein 1 (*Tp53inp1*) is a proapoptotic, stress-induced *p53* target gene that has the ability to interact with *p53* and modulates its transcriptional activity [52]. He et al. [53] found that the overexpression of miR-155 in acute myocardial infarction (AMI) induced cardiac fibrosis by directly targeting the *Tp53inp1* gene and inhibiting its expression, while downregulated *Tp53inp1* dramatically promoted the mRNA and protein expression levels of collagen I/III and increased the expression level of α-SMA in cardiac fibroblasts. We speculated that the upregulated miR-339-5p inhibited the expression of *Tp53inp1* in injured urethral tissues, resulting in the deposition of fibrin in urethral tissue and promoting urethral remodeling.

Duan et al. [54] demonstrated that the overexpression of *Dab2ip* can inhibit the proliferation, migration, and apoptosis of pancreatic cancer cells. In our results, *Dab2ip* was targeted by one upregulated miRNA, mir-31a-5p (Figure 5a,c). We speculate that the upregulated mir-31a-5p in injured tissues targets *Dab2ip*, resulting in the decrease in *Dab2ip* expression and promoting the proliferation and migration of urethral epithelial cells.

*Tp53inp1* and *Dab2ip* are tumor suppressor genes [52,54]. According to the GO and KEGG function annotation of *Tp53inp1* and *Dab2ip*, we found that they are involved in the negative regulation of cell population promotion, positive regulation of the apoptotic process, negative regulation of fibroblast promotion, and positive regulation of the apoptotic signaling pathway. In injured urethral tissues, we found that they were targeted by one upregulated miRNA, mir-339-5p and mir-31a-5p, respectively, and their expression decreased, resulting in the opposite effect. We can speculate that the inhibitory effect of related tumor suppressor genes is weakened after urethral injury.

In our study, mir-486 was found to be downregulated after urethral injury. We further found that mir-486 targeted *Wdr35*, *Radil*, *Csdc2*, and *Galnt11*, respectively, and they were all upregulated (Figure 5b,d). Some of these genes were annotated with GO and KEGG functional analysis, and we found that they were involved in positive regulation of the apoptotic process, positive regulation of the release of cytochrome c from mitochondria, substrate adhesion-dependent cell spreading, transcription factor binding, regulation of mRNA stability, metabolism of proteins, and the regulation of Notch signaling pathway. Previous studies have shown that upregulated mir-486 plays a protective role in PM2.5-induced human lung alveolar epithelial A549 cells by reducing cell apoptosis, preventing ROS production, and reducing cell injury [55]. Moreover, upregulated mir-486-5p inhibited the hyperproliferation and excessive production of collagen in hypertrophic scar fibroblasts via the IGF1/PI3K/AKT pathway [56]. Various studies suggest that mir-486 is a protective miRNA. Altogether, we speculate that the mir-486 can be used as a biomarker for the early diagnosis of urethral injury, and the regulation of mir-486 may be a potential intervention target in injured urethral tissues.

Despite this potential, there are several limitations of this study that should be acknowledged. First, although using animal models is a powerful tool for obtaining valuable bioinformatics evidence, the results cannot be extrapolated directly to humans because of species differences. Second, in addition to miRNAs, other non-coding RNAs, such as long non-coding RNAs (lncRNAs) and circular RNAs (circRNAs), also play important roles in the process of gene regulation induced by injury, which is worthy of comprehensive exploration in this regard. Finally, our results provide a number of candidate biomarkers or intervention targets; however, these candidates can only be predicted using available informatics analyses or literature, so further research is needed to validate their specific roles.

## 5. Conclusions

We provide the first identification of DE mRNAs and miRNAs in the urethra of rats with simulated clinical urethral injury using RNA sequencing and small RNA sequencing. Moreover, bioinformatics analysis identified promising candidate target genes and miRNAs and the target regulatory network were constructed. These findings provide valuable insights into the molecular mechanism of urethral injury and open up new possibilities for the development of novel therapeutic strategies for effective treatments.

## Figures and Tables

**Figure 1 genes-13-00824-f001:**
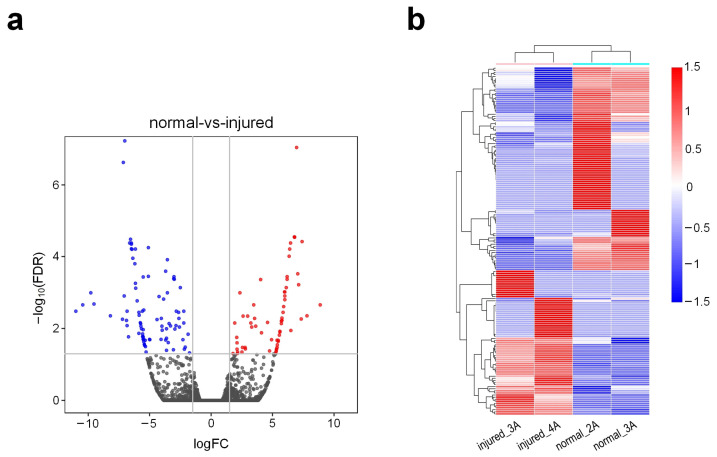
Expression patterns of genes in different groups of rat urethra. (**a**) Deregulated genes in injured tissues when compared to those in normal tissues. Red dots indicate the upregulated genes, blue dots indicate the downregulated genes and black dots indicate the non-differential genes (|log_2_(Fold Change (FC))| ≥ 1.5 and FDR value < 0.05); (**b**) The bi-clustering of genes expression profiles in different groups of rat urethra samples. The log_2_(FPKM + 1) of the transcripts were used in the analysis.

**Figure 2 genes-13-00824-f002:**
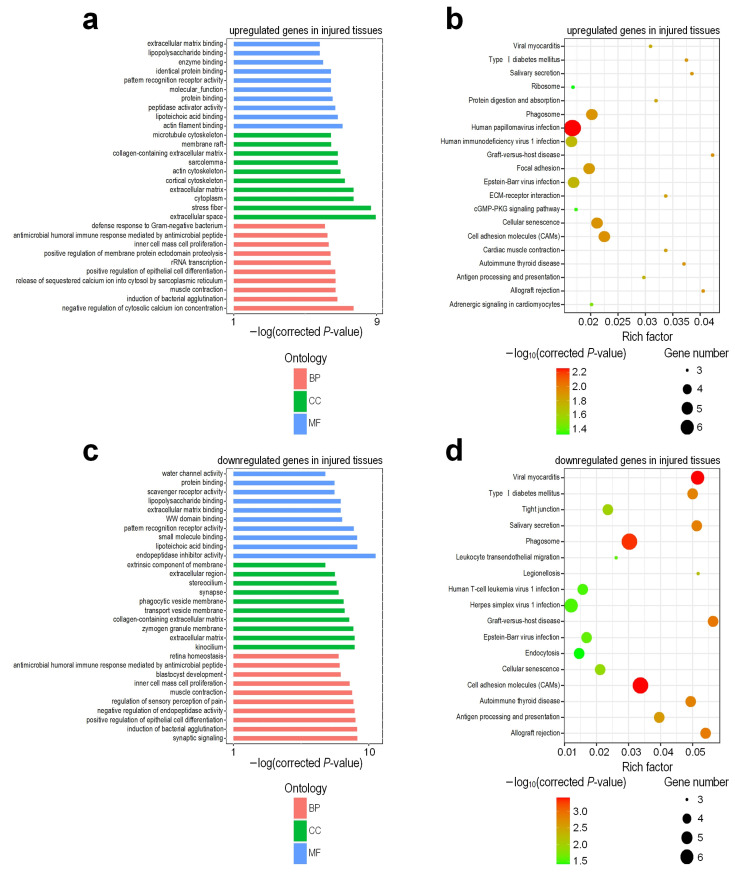
Enriched GO terms and KEGG pathways in deregulated genes when comparing RNA-seq profiles of the injured and normal tissues of rat urethra. (**a**) GO terms of upregulated genes in injured tissues; (**b**) Enriched KEGG pathways of upregulated genes in injured tissues; (**c**) GO terms of downregulated genes in injured tissues; (**d**) Enriched KEGG pathways of downregulated genes in injured tissues. In Part (**a**) and (**c**), the BP, CC and MF represent biological process, cellular component and molecular function, respectively. In Part (**b**) and (**d**), the Rich factor is calculated by dividing the number of input genes with the KEGG pathway by the total number of genes within the same pathway.

**Figure 3 genes-13-00824-f003:**
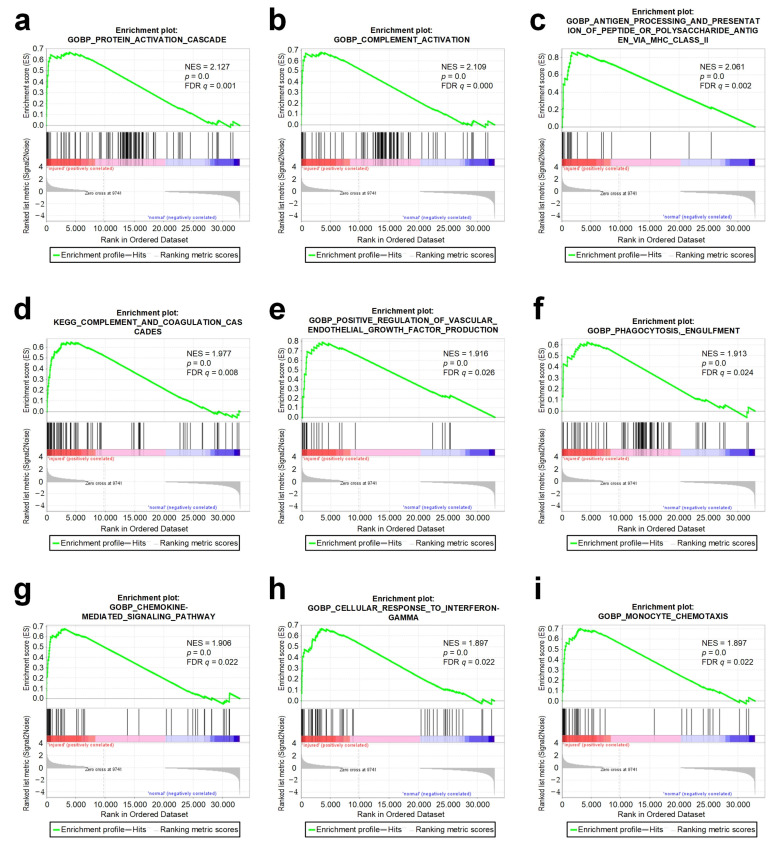
GSEA analysis of nine representative significantly enriched gene sets in phenotype injured. Enrichment plots comparing injured toward normal phenotype were depicted with following sets of genes: (**a**) GOBP protein activation cascade; (**b**) GOBP complement activation; (**c**) GOBP antigen processing and presentation of peptide or polysaccharide antigen via MHC class II; (**d**) KEGG complement and coagulation cascades; (**e**) GOBP positive regulation of vascular endothelial growth factor production; (**f**) GOBP phagocytosis, engulfment; (**g**) GOBP chemokine-mediated signaling pathway; (**h**) GOBP cellular response to interferon-γ and (**i**) GOBP monocyte chemotaxis. Comparison of samples, NES, nominal *p*-value, and FDR *q*-value were determined by the GSEA software and were indicated within each enrichment plot.

**Figure 4 genes-13-00824-f004:**
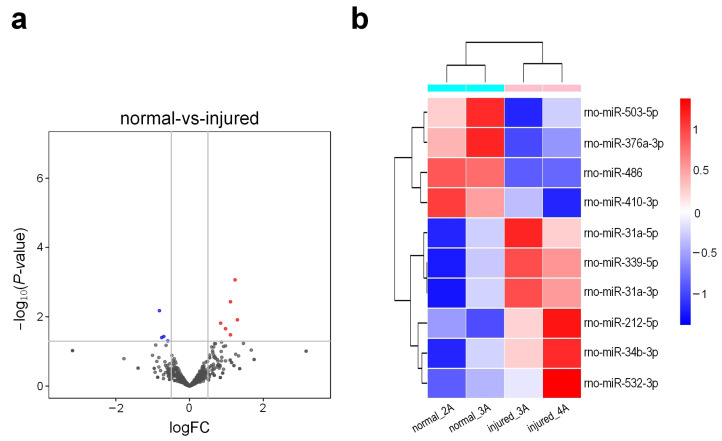
Expression patterns of miRNAs in different groups of rat urethra. (**a**) Deregulated miRNAs when comparing their expression levels in injured tissues to those in normal tissues. Red dots indicate the upregulated miRNAs, blue dots indicate the downregulated miRNAs and black dots indicate the non-differential miRNAs (|log_2_(Fold Change (FC))| ≥ 0.5 and *p*-value < 0.05); (**b**) The bi-clustering of miRNAs expression profiles in different groups of rat urethra in sRNA-seq profiles. The values shown are the log_2_(RPTM + 1) of the miRNAs.

**Figure 5 genes-13-00824-f005:**
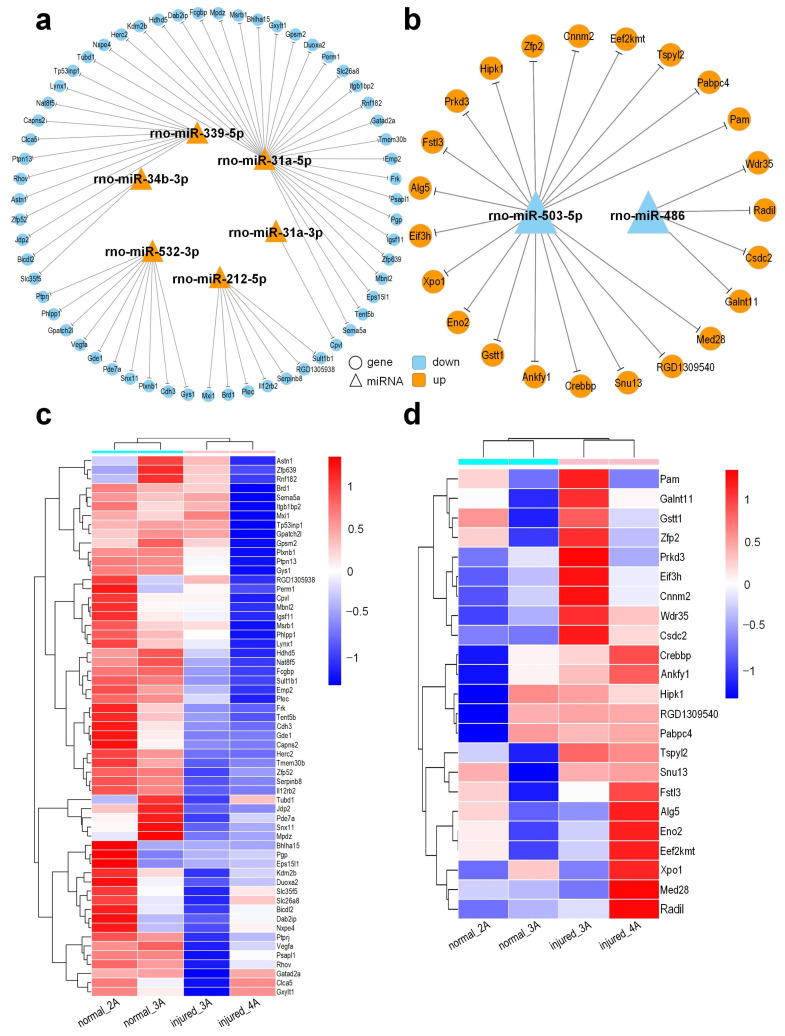
Differentially expressed miRNAs (DEMs) and their target genes in injured urethral tissues compared to normal urethral tissues. (**a**) The upregulated miRNAs and their target genes that were downregulated (logFC < −0.19) in the injured urethral tissues; (**b**) The downregulated miRNAs and their target genes that were upregulated (logFC > 0.1) in the injured urethral tissues. The triangles and circles represent miRNAs and target genes, respectively. The orange and blue colors represent upregulation and downregulation, respectively; (**c**) The bi-clustering of targets of upregulated miRNAs in Part (**a**) in different groups of rat urethra in sRNA-seq profiles; (**d**) The bi-clustering of targets of downregulated miRNAs in Part (**b**) in different groups of rat urethra in sRNA-seq profiles. In Part (**b**) and (**d**), the values shown are the log_2_(FPKM + 1) of the genes.

**Figure 6 genes-13-00824-f006:**
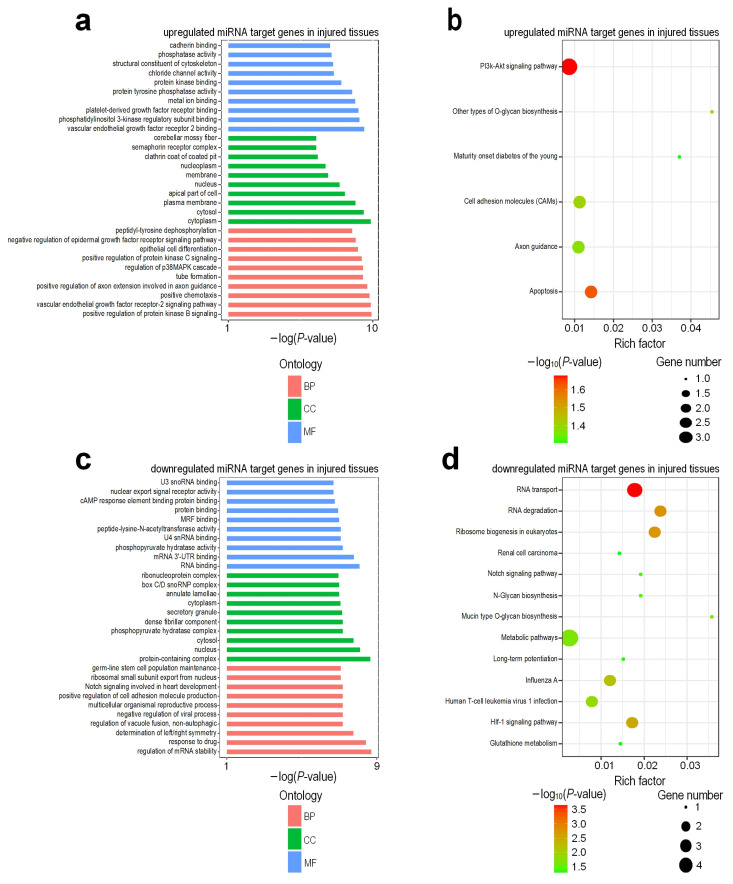
Enriched GO terms and KEGG pathways in deregulated miRNA target genes when comparing sRNA-seq profiles of the injured and normal tissues of rat urethra. (**a**) GO terms of upregulated miRNA target genes in injured tissues; (**b**) Enriched KEGG pathways of upregulated miRNA target genes in injured tissues; (**c**) GO terms of downregulated miRNA target genes in injured tissues; (**d**) Enriched KEGG pathways of downregulated miRNA target genes in injured tissues. In Part (**b**) and (**d**), the Rich factor is calculated by dividing the number of input genes with the KEGG pathway by the total number of genes within the same pathway.

## Data Availability

The RNA and sRNA sequence data have been submitted to the NCBI GEO database under the accession number GSE182642 and GSE182947, respectively.

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
