# Peer review of "Exploring Relevant mRNAs and miRNAs in Injured Urethral Tissues of Rats with High-Throughput Sequencing"

_genes, 2022, doi:10.3390/genes13050824_

Round 1

Reviewer 1 Report

In the current study, the authors have explored changes in mRNAs and miRNAs in injured urethral tissues of rats with high throughput sequencing. Differential expression of mRNAs and miRNA post clinical urethral injury were identified.

The article is well structured into sections and subsections and professionally written. It is within the scope of the journal.

However, there are some concerns that need to be addressed:

  • Could authors explain the rationale of using just two animals in each group (control vs injured)?

For the results to be conclusive usually n=10 in each group is recommended if not then at least n=3 should be considered.

  • Page 5, Line 221: The sentence needs rephrasing for clarity.
  • Page 7, Figure 2 and Page 14, Figure 6: There is a typing error; upregulated and downregulated spelling needs to be corrected.
  • Page 17, Line 518-533: The spelling of urethral needs to be corrected in multiple places.
  • Page 19: Check reference 34 and 37 for details and consistency.

Reviewer 2 Report

In this study, the authors comprehensively investigated the molecular mechanism of urethral injury via RNA-seq profiling of normal and injured urethral tissues to characterize DE mRNAs and miRNAs. They identified 166 DE mRNAs and 10 DE miRNAs. The study is interesting, the methodology is accurate, and the conclusion is supported by the results. The limitations are well recognized by the authors. However, I have a few minor issues that should be addressed as follows:

  • There are several structural, grammatical, punctuation, and capitalization errors that make the manuscript hard to read. Therefore, the manuscript must be revised by a native English speaker.
  • The epidemiological data regarding urethral stricture in China should be supported by a reference. (Lines 46-48).
  • "2.2. Establishment of Urethral Injury Animal Model": This section must be written using the past tense for parts from 119 to 123.
  • Supplementary figures should be arranged in sequential order as they appear in the text.

Reviewer 3 Report

The reviewed manuscript entitled ‘Exploring relevant mRNAs and miRNAs in injured urethral tissues of rats with high-throughput sequencing’ written by Han Lin et al. presents interesting insight into differentially expressed mRNAs and miRNAs in injured urethral tissues using in vivo models and next generation sequencing. The reviewed manuscript is properly designed and methodologically sound with informative visualisations. The paper is well organized and the English language is correct. My suggestions to the authors are addressed in the comments below.

General concept comments:

  1. On the heatmap in Figure 1, the expression of selected genes was visualized in two samples of injured tissue and two samples of normal tissue. For some genes, clear variability within studied groups is visible, e.g. some genes are highly expressed in injured_3A sample, while low expressed in injured_4A sample, and vice versa. Similar situation is also in normal samples group. Only genes clustered at the top and bottom of the heatmap appear to be uniformly expressed in both studied groups. Did the authors consider addressing of this bias, e.g. by excluding genes with the highest variability in expression within injured or normal groups?
  2. My second remark regards the prediction of genes regulated by obtained 10 miRNAs. If I understand well, genes regulated by these miRNAs were computationally predicted and their expression in the analyzed samples along with functional analysis was presented. However, I wonder if the better option is to select and show only these targets, which are included in the set of 166 genes selected from differential gene expression analysis, where stricter selection criteria were used. It could make the constructed miRNA-mRNA regulatory network more relevant and confident.

Specific comments:

  1. Line 52: The word ‘increases’ seems not to be necessary.
  2. Titles of four panels in Figure 2 contain typos in ‘uprugulated’ and ‘downrugulated’.

I believe that my suggestions will be helpful to the authors in increasing the quality of the reviewed article.
